

# Superoxide dismutase activity in tear fluid and blood of patients and mouse model of amyotrophic lateral sclerosis: a pilot study

Tatiana A. Pavlenko[1], Natalia B. Chesnokova[1], Olga V. Beznos[1], Natalia N. Shikareva[1], Marina R. Nodel[2], Ksenia V. Shevtsova[2], Uliana V. Panina[2], Daniil A. Shteinberg[3], Olga A. Kukharskaya[3,4], Iuliia S. Sukhanova[3,4], Nadezhda E. Pukaeva[3,4], Michail S. Kukharsky[3,4] and Ruslan K. Ovchinnikov[3,4]

[1] Helmholtz National Medical Research Center for Eye Diseases of the Ministry of Health of the Russian Federation, Moscow, Russia
[2] N. V. Sklifosovskiy Institute of Clinical Medicine, Moscow State Medical University (Sechenov University), Moscow, Russia
[3] Institute of Physiologically Active Compounds at Federal Research Center of Problems of Chemical Physics and Medicinal Chemistry, Russian Academy of Sciences, Chernogolovka, Russia
[4] Department of General and Cell Biology, Faculty of Medical Biology, Pirogov Russian National Research Medical University, Moscow, Russia

Corresponding author
Ruslan K. Ovchinnikov,
rusovc@mail.ru

## ABSTRACT

Amyotrophic lateral sclerosis (ALS) is a severe neurodegenerative disease characterized by progressive degeneration of motor neurons and skeletal muscle atrophy. The heterogeneity of clinical symptoms and the lack of reliable biomarkers hamper diagnostics of ALS. The dysfunction of superoxide dismutase 1 (SOD1) protein is considered one of the molecular mechanisms underlying ALS pathology. We measured total SOD activity in the tear fluid and blood serum of ALS patients, healthy volunteers, and in the ALS mouse model, harboring the human truncated form of fused in sarcoma (FUS) protein—FUS (1–359). The average SOD activity in tear fluid did not differ between ALS patients and the control group. However, an increased proportion of patients with low SOD activity in tear fluid was observed compared to the control group. In contrast, SOD activity in blood serum was higher in the ALS group. Transgenic FUS (1–359) mice showed decreased SOD activity in tear fluid at both presymptomatic and symptomatic stages of ALS. SOD activity in blood serum did not differ between transgenic and control animals. These findings suggest that changes in SOD activity in the tear fluid of ALS patients and transgenic FUS (1–359) mice reflect local metabolic disturbances in the eyes associated with ALS.

## INTRODUCTION

Amyotrophic lateral sclerosis (ALS) belongs to the group of motor neuron diseases characterized by motor neuron degeneration in the primary motor cortex, brainstem, and

spinal cord. This degeneration leads to paralysis and muscle atrophy. The average incidence is 2.2 per 100,000 people yearly (*Longinetti & Fang, 2019*), with an estimated survival of 2 to 5 years after symptoms onset. At present, this disease is considered incurable. In most cases (90–95%), ALS is sporadic. About 5–10% of ALS cases are hereditary (*Barberio et al., 2023*). Of them, 20% reveal mutations in the gene encoding Cu/Zn superoxide dismutase 1 (SOD1) (*Rosen et al., 1993*; *Skvortsova et al., 2001*). SOD1 mutations are also detected in sporadic ALS forms (*Andersen & Al-Chalabi, 2011*; *Muller et al., 2022*).

Neuronal cell death in ALS, similar to other neurodegenerative diseases, is associated with protein misfolding, endoplasmic reticulum stress, and mitochondrial dysfunction (*Mead et al., 2023*). Additionally, oxidative stress and neuroinflammation play a major role in the pathology of the disease (*Morgan & Orrell, 2016*; *Zhou & Xu, 2023*). By now, over 40 genes have been identified with mutations linked to hereditary forms of ALS (*Nguyen, Van Broeckhoven & van der Zee, 2018*). These mutations are mostly found in the following genes: C9ORF72 (*Renton et al., 2011*), SOD1 (*Rosen et al., 1993*), TARDBP (*Kabashi et al., 2008*), FUS (*Vance et al., 2009*), OPTN (*Maruyama et al., 2010*), PFN1 (*Yang et al., 2016*), VCP (*Johnson et al., 2010*), ANG (*Greenway et al., 2006*), TUBA4A (*Smith et al., 2014*). SOD1, one of the three SOD isoforms (SOD1, SOD2, and SOD3), is an antioxidant enzyme that catalyzes the dismutation of superoxide radicals into molecular oxygen and hydrogen peroxide (*Fukai & Ushio-Fukai, 2011*). Therefore, SOD1 protects cells against the oxidative stress. Reduced SOD1 activity leads to free radical accumulation, which causes nucleic acid damage and protein misfolding (*Barbosa et al., 2010*; *Chandimali et al., 2025*). Also, mutant forms of SOD1 exhibit an increased aggregation propensity (*Kaur, McKeown & Rashid, 2016*; *Pasinelli & Brown, 2006*). There is evidence of altered localization and accumulation of structurally disordered immature conformers of the SOD1 protein in spinal motor neurons across various forms of ALS (*Trist et al., 2022*). ALS development may also be associated with mutations in genes encoding DNA/RNA-binding proteins—FUS and TDP-43 (TAR DNA binding protein). In ALS patients with these mutations, FUS and TDP-43 become mislocalized from the nucleus to the cytoplasm forming aggregates in motor neurons (*Aksoy et al., 2020*; *Carey & Guo, 2022*; *Skvortsova et al., 2009*). TDP-43- and FUS-reactive inclusions are also detected in motor neurons of patients with sporadic ALS forms (*Ikenaka et al., 2020*; *Serdyuk, Levitsky & Skvortsova, 2006*; *Tyzack et al., 2019*). Previously, we generated a transgenic mouse line expressing a truncated form of human FUS (1–359) that induces FUS-proteinopathy accompanied by motor neuron death and ALS clinical patterns (*Shelkovnikova et al., 2013*).

In neurodegenerative diseases, such as Alzheimer's disease and Parkinson's disease, changes in the visual system occur, which could be explained by both the common embryonic origin of retinal ganglion cells and brain neurons and their interactions through neural and humoral mechanisms (*Chesnokova, Pavlenko & Ugrumov, 2017*; *Czako et al., 2020*; *Nowacka et al., 2014*; *Ornek, Dag & Ornek, 2015*; *Roda et al., 2020*; *Romaus-Sanjurjo et al., 2022*; *Zhang et al., 2021*). The ocular changes observed in ALS are primarily associated with the ocular motor system (*Aust et al., 2024*; *Cozza et al., 2021*; *Sharma et al., 2011*). However, visual acuity, contrast sensitivity, visual fields, and visually evoked potentials are minimally altered. At the same time, changes in color vision are noted in more than

half of ALS cases (*Boven, Jiang & Moss, 2017*). Histological examination of postmortem retinal samples and the *in vivo* optical coherence tomography revealed significant changes in the retina of ALS patients (*Rojas et al., 2020*; *Soldatov et al., 2021*). The increase in choroidal thickness correlates with neurodegenerative progression (*Cennamo et al., 2022*). Intraretinal protein inclusions (*Fawzi et al., 2014*; *Volpe et al., 2015*) and microglial activation (*Pediconi et al., 2023*) are detected in the retinas of ALS patients and transgenic ALS mice. These retinal changes may precede the onset of motor disturbances in ALS.

It is well known that changes in the composition of tear fluid occur during neurodegenerative processes in both the brain and the retina (*Ami et al., 2021*; *Gijs et al., 2021*; *Kallo et al., 2016*). This is caused by the complex neural and humoral regulation of tear production (*Chesnokova et al., 2023*). Changes in tear composition caused by Parkinson's and Alzheimer's diseases have been most studied (*Marchesi et al., 2021*; *Ohno et al., 2022*). Little is known about tear composition in ALS. Protein and lipid alterations and phenylalanine level reduction were found in the tears of ALS patients, which indicates metabolic changes in the tear fluid (*Ami et al., 2021*). SOD1 dysfunction and oxidative stress are considered key factors in ALS pathogenesis. Changes in SOD1 concentration in cerebrospinal fluid of ALS patients during treatment provided a basis for using SOD1 activity in cerebrospinal fluid as a biomarker of ALS progression and treatment effectiveness (*Gertsman et al., 2019*; *Winer et al., 2013*). However, there are contradictory data on how changes in SOD properties affect the total ability of SOD to interact with superoxide anion. Also, it is not clear whether SOD activity in tear fluid can be used as an indicator of ALS progression. Tear fluid collection is a minimally invasive procedure, making it a simple and fast method for assessing enzyme activity. This approach allows for longitudinal monitoring of SOD activity, potentially aiding in ALS research and diagnostics. In this study, we determined the total SOD activity in the tear fluid and blood serum of ALS patients and FUS (1–359) transgenic mice and demonstrated that alterations in SOD activity are present in both the patient cohort and the ALS animal model.

## MATERIALS AND METHODS

### Patients

The study was performed with samples obtained from 10 patients (20 tear fluid samples) with sporadic ALS—three males and seven females (Table 1). The patients were followed at the Neurology Department of Kozhevnikov Clinic of Nervous Diseases at University Clinical Hospital No. 3 of I.M. Sechenov First Moscow State Medical University. The ALS diagnosis was established based on clinical and electrophysiological confirmation of central and peripheral motor neuron involvement according to the diagnostic criteria of ALS (*Goutman et al., 2022*; *Johnsen, 2020*). ALS duration from the onset of neurological symptoms ranged at baseline from 1 month to 2 years. The bulbar region was affected in 50% of patients; the cervical and thoracic regions—in 20% of patients; and the lumbosacral region—in 40% of patients. The control group comprised 12 healthy volunteers (24 tear fluid samples) without neurodegenerative and eye diseases or serious comorbidities—three males and nine females. Written informed consent was obtained from all subjects involved
**Table 1  Socio-demographic and clinical characteristics of patients and control subjects.**

|  | ALS patients (n = 10) | Controls (n = 12) | Difference, p level |
|---|---|---|---|
| Age, years (±SD*) | 59,20 (±12,59) | 53,25 (±17,73) | 0,4063** |
| Gender, male/female | 3/7 | 3/9 | 1,0000*** |
| Age of onset ALS, years (±SD) | 58,7 (±13,07) | – | – |
| Duration of ALS, months (±SD) | 9,10 (±6,79) | – | – |
| Bulbar region affected, yes/no | 5/5 | – | – |
| Cervical and thoracic region affected , yes/no | 2/8 | – | – |
| Lumbosacral region affected, yes/no | 4/6 | – | – |

**Notes.**
*Standard deviation.
**Mann–Whitney test.
***Fisher exact test.

in the study and no private information identifying participants was made public. No additional samples for the purpose of this study were collected. Patients were informed of the research and their nonobjection approval was confirmed. The study was conducted according to the guidelines of the Declaration of Helsinki and approved by the Ethics Committee of the Federal State Autonomous Educational Institution of Higher Education I.M. Sechenov First Moscow State Medical University of the Ministry of Healthcare of the Russian Federation (protocol No. 1/10-2020, 09 December 2020).

Tear fluid from both eyes was collected in the morning before any medication was administered. Sterile strips of filter paper were placed under the lower eyelid for 5 min. The strips were then placed in a test tube with saline solution. After 20 min the strips were removed, and the tear eluate was used for further measurements. Blood was taken from the elbow vein and left at room temperature for 30 min until clot formation. The blood was then centrifuged at 2,000 g for 15 min at +4 °C, and the serum was collected.

## Animal model

The study was conducted with FUS (1–359) transgenic mice generated and maintained on the CD-1 genetic background as was described previously by our group (*Shelkovnikova et al., 2013*). These mice express a truncated form of a human FUS (1–359) protein, which leads to severe neurodegeneration with motor disturbances and animal death at 4–5 months of age. CD-1 mice were used as a wild type control. Animals were obtained from the Bioresource Collection of Institute of Physiologically Active Compounds at Federal Research Center of Problems of Chemical Physics and Medicinal Chemistry, Russian Academy of Sciences. All mice were genotyped by the PCR analysis of DNA obtained from ear biopsy, as was described previously (*Shelkovnikova et al., 2013*). The animals were used at the age of 6–8 weeks at the early stage of model disease development, when no signs of motor dysfunction were observed, and at the age of 12–15 weeks, when the first clinical signs of model disease began to appear. A total of 42 mice were divided randomly into four groups based on age and genotype, with 9–12 animals per group for tear fluid analysis and 5–9 animals per group for blood sampling. Sample sizes were estimated based on previous experience and calculation to achieve adequate power. The number of animals for each
experiment is provided in the figure legends. Experimental animals were maintained in standard conditions (22 ± 2 °C, 55–60% relative humidity) on a 12 h light/12 h dark cycle at no more than five adult animals per cage (365 × 207 × 140 mm), with food and water supplied *ad libitum*. Mice health status was checked daily. The procedures were performed in accordance with the "Guidelines for accommodation and care of animals. Species-specific provisions for laboratory rodents and rabbits" (GOST 33216-2014), in complience with the principles of the Directive "2010/63/EU" on the protection of animals used for scientific purposes. The procedures were also approved by the Ethics Reviewing Committee of the Institute of Physiologically Active Compounds at Federal Research Center of Problems of Chemical Physics and Medicinal Chemistry, Russian Academy of Sciences (protocol No. 53. 18 December 2023). Efforts were made to limit the number of animals in the experiment and to minimize their suffering. Mice displaying signs of distress, labored breathing, or significant weight loss were euthanized before the experiment concluded. Euthanasia of animals not needed for experiments was performed by carbon dioxide ($CO_2$) using a 30–70% per minute displacement of chamber air with compressed $CO_2$ immediately followed by cervical dislocation.

Tear fluid was collected from animals with sterile strips of filter paper (2.5 mm wide) and tear eluate was prepared as it was prepared from human tears. Tear fluid from both eyes was combined. A blood volume of 200 μl was collected from the retro-orbital sinus by using glass capillary tubes. Serum was then obtained with the same procedure as was described for patients. No adverse events were observed. Mice were anesthetized with 1.5% isoflurane (Laboratorios Karizoo S.A., Barcelona, Spain) in oxygen at 1 L/min (R500IE, RWD Life Science Co., Shenzhen, China). At the end of experiment, animals were euthanized as described above and tissues were collected and stored at −80 °C until use for Western blotting.

## Measurement of SOD activity

SOD activity was measured using a method based on SOD's ability to inhibit spontaneous oxidation of quercetin in the presence of TEMED at pH 10.2 (*Kostyuk, Potapovich & Kovaleva, 1990*). This method allows total activity evaluation of all isoforms of the SOD enzyme. The tear fluid eluate was added to the phosphate buffer (25 mM $KH_2PO_4$, 0.08 mM EDTA, 0.8 mM TEMED, Helicon, Russia) at pH 10.2 with 15,5 μM quercetin (Diam, Russia) solution in DMSO (Laverna, Russia). Then, absorbance at 406 nm was measured using a UV 160A spectrophotometer (Shimadzu, Japan). After 20 min of incubation at room temperature, absorbance was measured again. The difference between the first and second measurements for each sample was compared to the difference for reference samples (in triplicate) without tear fluid, and the percentage of inhibition was calculated using the following formula: (Δ Reference OD −Δ Sample OD/Δ Reference OD) × 100. SOD activity in the samples was determined using a calibration curve constructed with recombinant human Cu/Zn-SOD protein (Reksod, "Fermentative technology", Russia). SOD activity was expressed in units of activity per milliliter (U/ml). The total protein concentration in the tear fluid eluate was determined according to Lowry (*Lowry et al., 1951*).

### Western blotting

Tissues were homogenized directly in a loading buffer and denatured at 100 °C for 5 min. Proteins were separated by electrophoresis on a 10% sodium dodecyl sulfate-polyacrylamide gel (SDS-PAGE) and transferred to a polyvinylidene fluoride (PVDF) membrane (Hybon-P, Amersham, UK) using a semi-dry transfer method. The membrane was blocked in a solution of 4% non-fat dry milk in TBST buffer (Tris-buffered saline with 0.1% Tween 20) followed by incubation with primary anti-FUS (1:1000, Cat.N611385; BD Biosciences, Franklin Lakes, NJ, USA) and HRP-conjugated secondary (1:4000, Bio-Rad, Hercules, CA, USA) antibodies. Protein bands were visualized using enhanced chemiluminescence (ECL) detection (Thermo Fisher Scientific, Waltham, MA, USA), following the manufacturer's instructions. Membranes were reprobed with β-actin antibody (1:4000, Sigma-Aldrich, St. Louis, MO, USA) as a loading control.

### Statistical analysis

All graphs and statistical analysis were performed with GraphPad Prism 8 software (GraphPad Software, San Diego, CA, USA). Data are expressed as mean ± standard error. The number of human samples and animals per group, effect sizes for each comparison, and $p$-value cut-offs are provided in the figure legends where applicable. Effect sizes were calculated using Cohen's $d$ statistic. Experimenters were blinded to group assignments prior to the final analysis.

## RESULTS

### SOD activity is altered in ALS patients

As a pilot study, we evaluated the potential usefulness of total SOD activity in body fluids as a peripheral marker of ALS pathology. We measured SOD enzyme activity in the tear fluid and blood serum of a small cohort of ALS patients ($n = 10$) and healthy controls ($n = 12$). SOD activity in tear fluid varied widely in both groups (ALS: 411.5 ± 110.7 U/ml; control group: 442.3 ± 88.8 U/ml), with no significant differences between them (Fig. 1A). Notably, the difference in SOD activity in tears between paired eyes was significantly higher in the ALS group compared to the control group (Fig. 1B) suggesting possible asymmetric changes in SOD activity between the eyes. Surprisingly, SOD activity in the blood serum of ALS patients was 25% higher (118.1 ± 9.2 U/ml) compared to the control group (88.8 ± 13.1 U/ml) (Fig. 1C). No correlation was found between SOD activity in tear fluid and blood serum for ALS patients ($r^2 = 0.05$, $p = 0.53$, data not shown). Therefore, SOD activity is likely regulated in blood and tear fluid independently.

Next, we ranked SOD activity results from tear samples into three categories for detecting enzyme activity: low activity (below 100 U/ml), moderate activity (100–500 U/ml), and high activity (above 500 U/ml). The proportion of samples with low SOD activity was higher (40%) in the ALS group than in the control group (21%). The proportion of samples with high SOD activity was lower (25%) in the ALS group compared to the control group (38%) (Fig. 2).

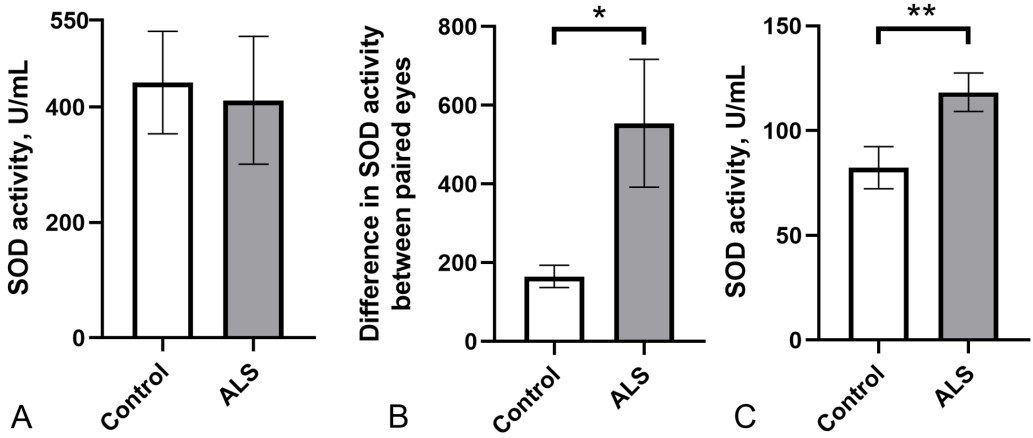

**Figure 1** **SOD activity in biological fluids of the ALS patients and the control group.** (A) SOD activity in tear fluid. (B) Asymmetry of SOD activity in tear fluid of paired eyes (difference in activity values between paired eyes). Mann–Whitney U test one-tailed, * - $p < 0.05$, effect size = 1.43. (C) SOD activity in blood serum. Mann–Whitney U test, ** - $p < 0.01$, effect size = 1.109; Control $n = 12$, ALS $n = 10$.

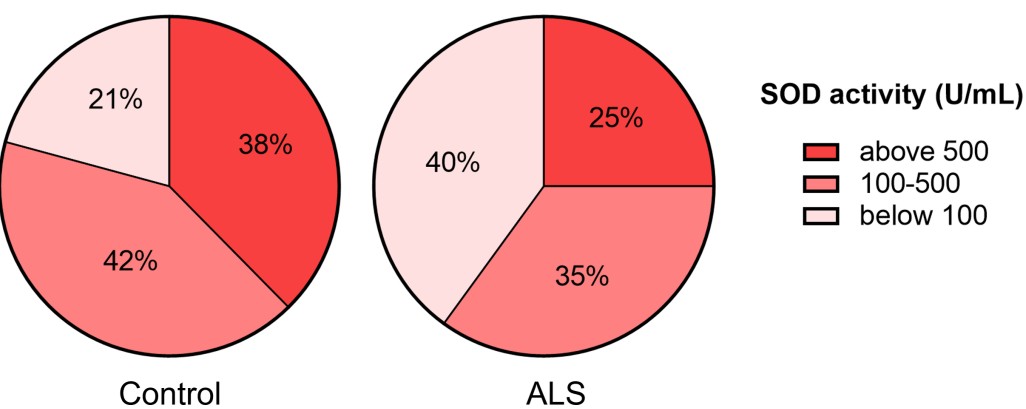

**Figure 2** **Distribution of tear fluid samples based on the ranges of SOD activity of the ALS patients and the control group.**

## SOD activity in transgenic FUS (1–359) mice

To test the hypothesis that ALS pathology associated with altered SOD activity we next measured SOD activity in the tear fluid and serum of transgenic mice FUS (1–359) which model ALS with pathology not linked to *SOD1* but to the mutated *FUS* gene, also genetically associated with ALS.

At the presymptomatic disease stage (age of 6–8 weeks), transgenic FUS (1–359) mice did not demonstrate any noticeable disturbances of motor function. The first symptoms such as limb paresis and paralysis usually appear at the symptomatic stage by the age of 12–15 weeks. These age periods were chosen to determine SOD activity in tear fluid and blood serum of these mice. No age-related changes in SOD activity in tears were found in both control and transgenic mice (Fig. 3A). However, in mice with FUS pathology, SOD

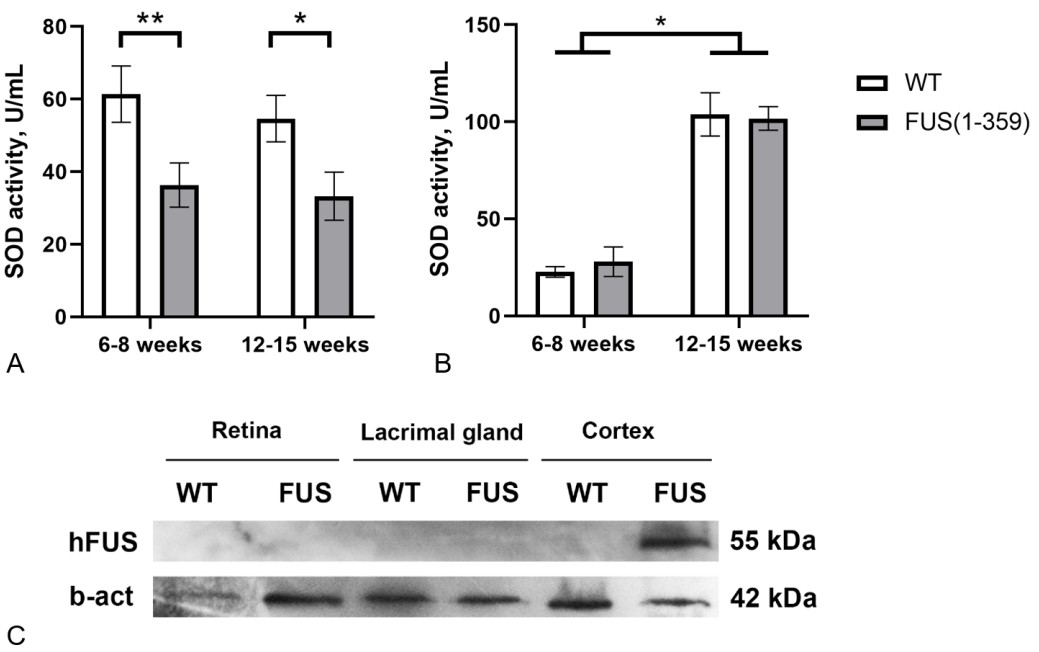

**Figure 3 SOD activity in FUS (1–359) transgenic mice and the wild type control at the presymptomatic (6-8 weeks) and symptomatic (12–15 weeks) stages of disease.** (A) SOD activity in tear fluid. Two-way ANOVA, Fisher's LSD test, *$p < 0.05$, effect size = 1.06, **$p < 0.01$, effect size = 1.07; 6–8 weeks WT $n = 11$, FUS (1–359) $n = 12$; 12-15 weeks WT $n = 10$, FUS (1–359) $n = 9$. (B) SOD activity in blood serum. Two-way ANOVA, Fisher's LSD test, *$p < 0.05$, effect size = 4.24 for WT; 6–8 weeks WT $n = 7$, FUS (1–359) $n = 5$; 12–15 weeks WT $n = 6$, FUS (1–359) $n = 9$. (C) Human FUS (1–359) protein detected only in the neural tissues of transgenic (FUS) but not retina, lacrimal gland or wild type (WT) samples using an antibody against the FUS protein.

activity in tears was significantly lower (on average by 40%) than in control animals. The decreased SOD activity was observed at both presymptomatic and symptomatic stages. Age-dependent increase in SOD activity was observed in blood serum, with no differences between the groups at both age periods (Fig. 3B). To exclude the possibility of a direct impact of pathogenic FUS (1–359) aggregates on eye physiology or tear production, we assessed FUS protein in the retina and lacrimal gland by western blotting. As expected mutated FUS was detected in brain but not in retina and lacrimal gland of transgenic mice (Fig. 3C).

## DISCUSSION

Several types of SOD constitute a group of metalloproteins catalyzing the dismutation of superoxide anion radicals into hydrogen peroxide and oxygen. SOD has the highest known catalytic rate constant ($\sim109$ M$-1$ s$-1$) and plays a key role in the body's antioxidant defense (*Miller, 2012*). Cytosols of almost all eukaryotic cells contain SOD1. Mitochondria of eukaryotic cells and many bacteria contain SOD2, whereas SOD3 is an extracellular form of the enzyme (*Perry et al., 2010*).
The important questions are whether there are any changes in the total enzymatic activity of SOD in biological fluids of ALS patients and whether they reflect the course of the disease and can be used for prognosis. Several studies have been devoted to these questions. SOD activity was measured in erythrocyte lysates, cerebrospinal fluid, brain structures, and blood plasma. SOD activity in blood during ALS was mostly studied in erythrocyte lysates that contain SOD1. Some studies reported decreases in SOD1 activity in erythrocytes of patients with familial ALS (*Bowling et al., 1995*; *Przedborski et al., 1996*). This abnormality of SOD1 activity can be detected years before the onset of ALS clinical manifestations in SOD1 mutation carriers (*Robberecht et al., 1994*). There is also evidence of decreased SOD activity in erythrocytes in sporadic ALS (*Apostolski et al., 1998*; *Nikolic-Kokic et al., 2006*). However, other studies did not find changes in specific SOD1 activity in sporadic ALS (*Puymirat et al., 1994*). *Ihara et al. (2005)* found a decrease in SOD activity in erythrocytes and the cerebrospinal fluid of patients with both familial and sporadic forms of ALS. At the same time, a significant increase in SOD1 activity in erythrocytes was reported in patients with sporadic ALS without pyramidal signs, who had slow ALS progression (*Bonnefont-Rousselot et al., 2000*). The authors conclude that high levels of enzyme activity are associated with mild disease progression (*Ihara et al., 1995*). *Moumen et al. (1997)* reported increased SOD activity in the blood plasma of patients with sporadic ALS. The authors suggest that this increase may reflect the involvement of extracellular SOD (SOD3). Contrary, according to *Cohen et al. (1996)* SOD activity is decreased in the blood serum of patients with sporadic ALS. These contradictions in the results regarding SOD activity in the blood of ALS patients can be attributed to differences in measurement methods, as well as variations in age groups and ALS etiology.

We assessed SOD activity in the tear fluid and blood serum of ALS patients and healthy volunteers. Notably, SOD activity in the tear fluid was 4–5 times higher compared to the SOD activity in the blood serum in both groups. We speculate that high SOD activity in tears is needed for intensive antioxidant protection because the eye surface is in contact with atmospheric oxygen and is subject to ultraviolet radiation. Although the differences in SOD activity in tear fluid between the ALS and control groups were not found, a shift towards a low range of activity was noted in the ALS group (Figs. 1 and 2). This may indicate that in a subset of ALS patients, the disease may be associated with impaired SOD function that also affects the eyes. Further studies with a larger cohort of ALS patients are required to confirm this and to explore potential genetic factors, such as *SOD1* mutations, that may influence SOD activity in tear fluid. In contrast to tear fluid, SOD activity in the blood serum was 30% higher in ALS patients compared to the control group (Fig. 1). The increased SOD activity in blood serum might indicate a systemic compensatory response against oxidative stress accompanying ALS pathology, especially in the early stages of the disease. The decreased SOD activity in tear fluid demonstrates local changes in the eye, which is isolated from the blood circulatory system.

Next, we measured total SOD activity in the tear fluid and blood serum of FUS (1–359) transgenic mice. This mouse model reproduces key phenotype features of ALS including death of motor neurons and progressive deterioration of motor functions (*Shelkovnikova et al., 2013*). We found that SOD activity in the tear fluid of FUS (1–359) mice was almost

twice as low compared to control mice at the early asymptomatic ALS stage (Fig. 3A). This difference persisted at the symptomatic stage, suggesting that impaired SOD activity in the tear fluid may be considered as a predictive marker across different stages of the disease. Notably, SOD activity in blood serum was the same in transgenic and wild type mice, and the overall SOD activity increased with age. A possible explanation for the differences in SOD activity between tissues or across different ages is that SOD protein may have different turnover rates (*Crisp et al., 2015*). To sum up, FUS pathology in FUS (1–359) mice affects SOD activity in tear fluid but not in blood, emphasizing the potential of tear fluid as a valuable source of biomarkers for neurodegenerative diseases.

## CONCLUSIONS

A higher percentage of ALS patients with low SOD activity in the tear fluid, significant differences in SOD activity in the paired eyes, and low SOD activity in the tear fluid of ALS transgenic mice indicate local metabolic eye changes in ALS. Therefore, assessing SOD activity in tear fluid could be used as a new minimally invasive method for improving diagnostics of ALS, including its FUS-associated forms. This would also assist in identifying risk groups among patients.

## ACKNOWLEDGEMENTS

Animals were provided and supported by Bioresource Collection of IPAC RAS and Centre for Collective Use IPAC RAS facilities and equipment in the framework of the State Assignment of IPAC RAS (FFSG-2024-0021, FFSG-2024-0023).

### Funding

This work was funded by the Russian Science Foundation (grant no. 22-75-00112). Animal samples analysis was supported by PRNRMU grant to R.K.O no. 205. The funders had no role in study design, data collection and analysis, decision to publish, or preparation of the manuscript.

### Grant Disclosures

The following grant information was disclosed by the authors:
The Russian Science Foundation: 22-75-00112.
PRNRMU: R.K.O no. 205.

### Competing Interests

The authors declare there are no competing interests.

### Author Contributions

- Tatiana A. Pavlenko conceived and designed the experiments, prepared figures and/or tables, authored or reviewed drafts of the article, and approved the final draft.

- Natalia B. Chesnokova conceived and designed the experiments, authored or reviewed drafts of the article, and approved the final draft.
- Olga V. Beznos performed the experiments, prepared figures and/or tables, and approved the final draft.
- Natalia N. Shikareva analyzed the data, prepared figures and/or tables, authored or reviewed drafts of the article, and approved the final draft.
- Marina R. Nodel analyzed the data, prepared figures and/or tables, and approved the final draft.
- Ksenia V. Shevtsova analyzed the data, prepared figures and/or tables, and approved the final draft.
- Uliana V. Panina analyzed the data, prepared figures and/or tables, and approved the final draft.
- Daniil A. Shteinberg performed the experiments, prepared figures and/or tables, and approved the final draft.
- Olga A. Kukharskaya performed the experiments, prepared figures and/or tables, and approved the final draft.
- Iuliia S. Sukhanova analyzed the data, prepared figures and/or tables, and approved the final draft.
- Nadezhda E. Pukaeva performed the experiments, prepared figures and/or tables, and approved the final draft.
- Michail S. Kukharsky conceived and designed the experiments, authored or reviewed drafts of the article, and approved the final draft.
- Ruslan K. Ovchinnikov conceived and designed the experiments, authored or reviewed drafts of the article, and approved the final draft.

## Human Ethics

The following information was supplied relating to ethical approvals (*i.e.*, approving body and any reference numbers):

The study was conducted according to the guidelines of the Declaration of Helsinki and approved by the Ethics Committee of the Federal State Autonomous Educational Institution of Higher Education I.M. Sechenov First Moscow State Medical University of the Ministry of Healthcare of the Russian Federation (protocol No. 1/10-2020, 09 December 2020).

## Animal Ethics

The following information was supplied relating to ethical approvals (*i.e.*, approving body and any reference numbers):

The procedures were also approved by the Ethics Reviewing Committee of the Institute of Physiologically Active Compounds at Federal Research Center of Problems of Chemical Physics and Medicinal Chemistry, Russian Academy of Sciences (protocol No. 53, 18 December 2023).

## Data Availability

Raw data is available in the Supplemental Files.

## Supplemental Information

Supplemental information for this article can be found online at http://dx.doi.org/10.7717/peerj.19623#supplemental-information.

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
