# Peer review of "Superoxide dismutase activity in tear fluid and blood of patients and mouse model of amyotrophic lateral sclerosis: a pilot study"

_PeerJ, doi:10.7717/peerj.19623_

## Round 0.1 · original submission · Major Revisions

The reviewers appreciated the relevance of your study and its potential to advance the understanding of ALS biomarkers. However, they have identified areas that require attention before the manuscript can be considered for publication.

One of the main concerns relates to the methodology and sample sizes used in your study. The reviewers questioned whether the statistical power is adequate for the presented data, both in human and animal samples. They recommend clarifying and, where possible, expanding on these aspects to strengthen the robustness of your findings.

Additionally, the reviewers highlighted that certain sections of the manuscript, particularly the introduction and discussion, would benefit from greater precision and additional contextual support. There are instances where claims are not sufficiently referenced or where the discussion could delve deeper into the implications of your findings. This would help to better situate your study within the broader scientific literature and ensure your conclusions are well supported.

The presentation of data in the figures was also noted as an area for improvement. Specifically, the reviewers suggested clearer labeling and representation of statistical measures, as well as alternative ways to present certain data that might enhance clarity and interpretability for readers.

Finally, technical details related to your experimental design, including descriptions of controls, calibration methods, and certain reagents, were raised as points needing further elaboration. Addressing these questions will help ensure transparency and reproducibility of your study.

Reviewer 1 ·

Basic reporting

This manuscript examines SOD (all isoforms) activity in blood serum and tears from ALS patients and age matched controls and also in a FUS animal model. The authors main conclusion is that SOD activity is different in blood serum vs tears in ALS patients and also different SOD activity in tears and blood serum from a FUS animal model. The results support their conclusions as written, although one standard deviation from mean is quite large for some of their data (see Fig 1A, 1B). To further test the hypothesis, more mouse models of SOD should be examined and examined much longer than 12-15 weeks. Same with patients, most patients are less than 2 years of disease (if this were ever to be a marker of ALS prognosis). Organization of paper is good, may I suggest you add more references to your introduction, as there are some claims that are not cited.

Experimental design

The FUS mouse is on a CD-1 background (refs Delkin 2014, Shelkov..2013)…I looked up the Delkin 2014 reference and I cannot find this article in English. Does your control mouse in this study have human FUS wild type as control (not transgene), and is the control mouse at the same FUS expression or protein level as the human transgenic FUS under study? Or is this mouse control expressing mouse wild-type FUS? That is not clear to me. Can you give more information on your power analysis with 10.5 mice per group (4 groups, 42 total mice as mentioned in methods), such as what power level and what alpha level to derive those numbers of mice. Also, provide power analyses for your human samples (ie at power level of 0.8 we expect…..). What I’m getting at is do you have enough power here (says you do), but a sentence in the methods on this would be sufficient provided you tested enough numbers, especially the mice, seems small for some groups when I looked at your raw data.

To compare mice with humans, ideally you should see what age of mice (in months) is comparable to the age groups of your human controls and ALS patients. It seems your mice are younger here than the human groups, so aging them out longer and retesting them would be a better comparison to humans.
Your method on SOD activity (Kostyuk 1990) includes all SOD isoforms examined, and the calibration curve only uses recombinant SOD1 (not SOD2 or SOD3). Might be worth looking at recombinant SOD2 and SOD3 if you continue to use this approach, as you are measuring SOD activity of all isoforms. Perhaps just take recombinant SOD1, SOD2, SOD3, and run them in the calibration curve (ie could you just combine them at equal concentrations?). Or more tediously, just also do recombinant SOD1, SOD2, and SOD3 calibration curves separately and compare results.

Western blotting – is there an issue here with heating your samples at 100 degrees C? Such as could you be denaturing your protein and changing its shape? Were non-heated samples tried with a RIPA buffer and run on native PAGE to compare? Will you provide more information on the FUS antibody used. It seems to be a human FUS antibody recognizing all forms of FUS, but more details would be appreciated on this antibody from the Bioscience company (I tried to look it up this antibody and could not find it).

Statistical analysis – I recommend using standard deviation and not standard error of the mean for your figures. I calculated your mean and std deviation for all data provided, and your conclusions don’t change, but your std deviation has a lot more variation in Fig 1A, 1B. You should label mean plus/minus std deviation or std error of mean in your figure description. I think Figure 2 could be better represented by using bar graphs of mean plus/minus std deviation or std error of mean as you are using and compare them by your 3 groups for controls and ALS. That would make the figure more apparent, rather than looking at two pie charts, as it is not obvious what you are concluding.
Ideally, in the future, you should examine biofluids from disease control patients also, say Parkinson’s disease or other neuro diseases, in addition to age matched controls and ALS you describe in paper. That might provide insight into specificity of SOD activity for ALS versus other neurodegenerative diseases. If it will ever be used as a biomarker (diagnostic or prognostic), then you would want to test disease controls and know. I think commenting about this and perhaps mentioning more about the longitudinal investigation of SOD activity might provide more insight into this matter.

Validity of the findings

Why such large variation in the tears but not so much in the serum (see your error bars in Fig 1A and 1B, one standard deviation from the mean is 400+ units)? It might be worth to follow up on SOD1 protein levels in tears vs blood serum. I think a nano LC/mass spectrometer to quantify SOD1 protein level could do this (not an antibody), which you could just ship samples off to company to do this if not available to you.

How do the ages of your mice correspond to the ages in the human samples studied? I mentioned this already. These mice are younger overall relative to your human patients, I think. Might be worth mentioning in your discussion, and the limitations of the FUS model. Having a control mouse expressing wild type human FUS would be an ideal control to assess the impact with the human FUS (as written in your methods, it seems the control mouse is wild type mouse FUS, yes?). You might want to look at a knock-in mutation in a SOD mouse model that is not overexpressing a transgene (has this been done already, I don’t know, so if it has, then just ignore this).

I think it is worth talking about in your discussion that SOD1 protein may have different turnover rates in tissues (see “in vivo kinetic approach reveals slow SOD1 turnover in the CNS” – JCI, 2015, DOI: 10.1172/JCI80705), with the longest half-life of SOD1 protein in spinal fluid and the CNS (at least in rats in this paper). That could explain some of the SOD activity differences between tears (longer half-life I would presume) versus the blood serum in your animal studies. Not sure the half-life findings translate over the humans.

Additional comments

Some of this is redundant, as I came back and did a second review.

Abstract – line 28 and 29 – I think you mean ‘Cu/Zn’ SOD, and not just ‘SOD’. When I see ‘SOD’ in writing, that implies to me all SOD protein – SOD1, SOD2, SOD3.

Citations needed, line 48 – “SOD1 mutations are also detected in sporadic ALS forms”. You should reference citations for this statement. Please also add citations elsewhere in your introduction where needed, there are some claims made that have no references and these claims may not be general acceptance to those who study SOD.

The discussion of tears and blood serum SOD activity seems to have mixed results. Perhaps you could tighten up that paragraph, so it is not so verbose.

Spelling of ‘standard’ in Table 1 needs to be corrected

Fig 1 – should include text, “mean plus/minus standard deviation” As written, I think you used the standard error of the mean. Needs to be clearly stated what you used.

The pie charts are confusing to me, only one slice is really different between the two group comparisons, so perhaps highlight that more through a bar graph comparison of the 3 groups (ALS vs control), otherwise this figure seems like it just takes up space and doesn’t add much as it is just data from Fig 1A.

Reviewer 2 ·

Basic reporting

Dear authors and editors,
I agreed to review because the topic is very interesting and advancing biomarkers with diagnostic and/or progression value would represent a significant breakthrough for ALS patients. However, this paper has been disappointing me because:
1) The writing is unclear and not sufficiently supported by the available bibliographic references.
2) The methodology used is not the most appropriate, and the sample size (patients) is insufficient and overly heterogeneous.
3) The discussion is vague and poorly supported.

Experimental design

1. In the results section, you only present the SOD enzymatic activity. However, there is a figure showing western blot results for FUS determination in different samples i.e., retina, lacrimal gland and cortex. What is the purpose of this analysis? Just to prove that hFUS is only present in the cortex? Why is the procedure for obtaining the retinas not described in the Materials and Methods section?
2. Why did you use a method that only measures total SOD activity instead of differentiating the different isoforms (SOD1, SOD2 and the extracellular SOD3)? What about CSF?
3. The number of patients included in this study is too low. Moreover, the significance is very limited because:
• At no point is it mentioned whether familial or sporadic ALS cases carrying mutations affecting SOD were included or excluded from the study.
• The age range of patients included in the study is very broad, and based on the data provided, it is highly likely that both familial ALS cases, which typically have an earlier onset (58.7 - 13.07 ≈ 45 years), and sporadic ALS cases, which usually have a later onset, were included.
• The variability range in disease duration is also very wide, as the authors report it ranges from 9.10 - 6.79 = 2.31 months (very early ALS onset) to approx. 15 months of ALS (which in ALS patients means a long clinical evolution).

Validity of the findings

Regarding Results and Discussion
• The titles of figures 1 and 2 should mention that samples have been obtained from ALS patients. The legend of Figure 2 should include statistical analysis of data.
• Regarding Figure 3, SOD activity decreases in tear fluid, showing no changes in blood serum as the disease progresses when compared to WT controls. The authors should propose a hypothesis or justification in the discussion before claiming that SOD can serve as a disease progression marker. To obtain a more accurate explanation, a more appropriate methodology should be used (as noted in my previous comments), measuring the activity/levels of different isoforms of SOD rather than just the total activity. SOD1 and SOD2 could reflect tissue

Additional comments

Formal presentation and writing
1. Authors should improve the writing and avoid imprecise sentences: Abstract: “An increased proportion of patients with low SOD activity in tear fluid was observed” …but, compared with?
2. Bibliographic references must be introduced based on the publication date. I mean, inside the parenthesis, the oldest reference should be cited first.
3. In the abstract, contradictory messages are provided: “The average SOD activity in tear fluid did not differ between ALS patients and the control group” vs “These findings suggest that changes in SOD activity in the tear fluid of ALS patients reflect local metabolic disturbances in the eyes associated with ALS”.
4. General and imprecise sentences are provided, i.e.:
• Line 73: “In most cases, ALS is sporadic”. You must be precise in citing epidemiologic available data.
• Line 85: “Morphological changes should be preceded by metabolic changes that can be detected in biological fluids” You follow this sentence by citing examples in Alzheimer but not in ALS disease.

5. In the introduction:
• The existing literature on SOD expression and/or activity in tissues and samples from various ALS animal models, as well as in blood and CSF from patients, should at least be mentioned. The authors refer to it in the discussion, but not appropriately. It should be specified whether the statements refer to patients or animal models with genetically linked SOD ALS, or not.
• Contrary to what is stated in line 73: “In ALS patients, the main changes occur in the ocular motor system” (no reference is provided). You make mistakes, since the ocular motor system function is almost conserved in ALS patients.

---

## Round 0.2 · Minor Revisions

The authors have addressed all of the reviewers' comments. However, reviewer #1 has pointed some minor corrections still to be considered

Reviewer 1 ·

Basic reporting

no comment

Experimental design

CD-1 mice clarification (one more time) - the wildtype control mouse has a human transgene of full-length FUS or not? If not, you need to consider making a mouse with the full-length human FUS transgene as control (if control mouse does not have this) if you plan to do more studies with this mouse model of ALS. Not just a CD-1 mouse (mouse FUS only?).

Validity of the findings

no comment

Additional comments

My comments about things needing correction prior to publication (it would improve the paper in my opinion):

Line 48/49 - SOD1 mutations in sporadic ALS, either elaborate on this briefly or remove. If there are SOD1 mutations in sporadic ALS, then what does that imply if SOD1 mutations are thought to be mostly inherited? Are these individuals possessing SOD1 mutations with sporadic ALS have no family history with SOD1 mutations? How do you explain this?

Line 59 - remove 'toxic', SOD1 does dismutase function regardless of superoxide levels being toxic or not.

Line 61/62 - provide citations on nucleic acid damage and protein misfolding. That claim needs support.

Line 65 - correct to 'many forms of ALS' I don't think they looked at some other rare familial ALS cases (e.g. FUS mutations, etc.). Only C9ORF72-ALS, SOD1-ALS, and sporadic ALS I recall.

Line 71 - change 'earlier' to 'previously'. Reads better in my opinion

End of your introduction - important, you should write about the results and conclusions of your study here after line 105.

line 212/213 - "Therefore, SOD activity is likely regulated in blood and tear fluid independently". SOD (at least SOD1) is expressed in all tissue types. I'm providing this citation to help you think about what might be occurring in the periphery of ALS and what occurs in the ALS CNS. Might be helpful for thinking about what is going on with SOD activity in tears or blood. PMID: 33859434

line 243 - you need a citation here

line 264/265 - what about SOD isoforms (1,2,3) - are all these studies done by measuring activity of all SOD isoforms, or do they measure only SOD2, SOD1, etc.? Could that explain differences in these studies?

line 268 - write 'we speculate', no 'we suppose', reads better in my opinion.

Reviewer 2 ·

Basic reporting

Compared to the previous version, the authors have introduced the necessary improvements to enhance the presentation of the formal aspects of the work. The introduction, discussion, and figure titles have been improved, and relevant bibliographic references have been added.

Experimental design

Undoubtedly, the weak point of this article is the experimental methodology. It would have been highly advisable to assess the different SOD activities using a more appropriate methodology and to conduct a comparative study with values in cerebrospinal fluid. The number of patients is small, and the variability within each group is quite high. If this were not a rare disease like ALS, the article would likely have been rejected on this basis alone. I will accept the publication of the study because I understand how difficult it is to obtain a significant sample of ALS patients, and because this is intended as a 'proof of concept' that will require further studies.

Validity of the findings

The validity of the results is only relative based on what was stated in the previous section

---

## Round 0.3 · accepted · Accept

The authors have addressed all of the reviewers' comments and I believe the current version is ready for publication.